# Arctic Sea Ice Surface Roughness Estimated from Multi-Angular Reflectance Satellite Imagery

**Anne W. Nolin [1,\*] and Eugene Mar [2,†]**

1    Department of Geography, University of Nevada, Reno, NV 89557, USA
2    College of Earth, Ocean, and Atmospheric Sciences, Oregon State University, Corvallis, OR 97331-5503, USA;
     gmar31@comcast.net
*    Correspondence: anolin@unr.edu
†    Retired.

**Abstract:** Sea ice surface roughness affects ice-atmosphere interactions, serves as an indicator of ice age, shows patterns of ice convergence and divergence, affects the spatial extent of summer meltponds, and affects ice albedo. We have developed a method for mapping sea ice surface roughness using angular reflectance data from the Multi-angle Imaging SpectroRadiometer (MISR) and lidar-derived roughness measurements from the Airborne Topographic Mapper (ATM). Using an empirical data modeling approach, we derived estimates of Arctic sea ice roughness ranging from centimeters to decimeters within the MISR 275-m pixel size. Using independent ATM data for validation, we find that histograms of lidar and multi-angular roughness values were nearly identical for areas with a roughness < 20 cm, but for rougher regions, the MISR-estimated roughness had a narrower range of values than the ATM data. The algorithm was able to accurately identify areas that transition between smooth and rough ice. Because of its coarser spatial scale, MISR-estimated roughness data have a variance about half that of ATM roughness data.

**Keywords:** sea ice; surface roughness; remote sensing; MISR

## 1. Introduction

Sea ice roughness is created by surface-atmosphere interactions, ice motion, and ice surface melt. Roughness caused by ice motion depends on wind speed and direction, ocean currents, and coastline interactions [1]. Surface roughness is a characteristic of different sea ice types [2–4] and can serve as indicator of ice thickness, divergence, and convergence, all of which affect ship navigation in the polar regions [3]. Sea ice thickness is related to sea ice age where first-year ice is typically thinner (and smoother) than multi-year ice [5]. New ice types such as so-called grease ice, nilas, and pancake ice are relatively smooth, while pack ice, multi-year ice, and sea ice in the marginal ice zone tend to be rougher as they are affected by compression, shear, and wave action [5,6]. Ice divergence causes leads and polynyas, which are openings in the ice that expose ocean water to the atmosphere. Convergence causes ice ridges, which are very challenging for ship navigation.

Sea ice surface roughness is required for boundary layer climate modeling since it directly affects surface wind stress and determines aerodynamic roughness length [6–8]. Seasonal snow deposition on sea ice tends to smooth the surface and reduces the aerodynamic roughness length [9,10], while subsequent wind erosion and redeposition of blowing snow creates sastrugi and snow dunes on the ice [10,11]. Sea ice roughness enhances atmospheric boundary layer turbulence, thereby affecting turbulent energy transfer and boundary layer height [11–13].

There is a critically important relationship between winter and spring sea ice surface roughness, summer meltpond extent, and summer sea ice albedo. Crevices present in rough ice prior to the

melt season can persist, and when meltponds form, the meltwater tends to be spatially confined. This is in contrast to smoother ice that allows meltwater to spread laterally. Because meltponds on sea ice have a much lower albedo than non-melting sea ice (0.2–0.4 compared with 0.6–0.65, respectively) [14,15], surface roughness is a strong contributor to meltpond area [16]. As the areal fraction of meltponds increases, there is a linear decrease in sea ice albedo [16] and a strong ice-albedo positive feedback [15,17,18]. Late winter sea ice roughness has been shown to account for 85% of the variance in late summer ice albedo [18]. Thus, roughness is an important diagnostic indicator of sea ice processes and a valuable prognostic indicator of Arctic sea ice albedo.

Remote sensing is a valuable tool for assessing the state of sea ice over space and time. However, sea ice surface roughness is a source of uncertainty in remote sensing and itself has been a challenging parameter to retrieve. Extending from 1978 to the present, satellite remote sensing of sea ice is one of the longest continuous remote sensing records available [19], but surface topography measurements (e.g., radar, scatterometry, laser altimeter) have been too coarse to adequately characterize spatial and temporal changes in sea ice roughness. Airborne lidar data over sea ice provides fine scale spatial resolution data over sea ice, but the overall spatial and temporal coverage is sparse.

In previous work, Nolin et al. [20] described initial success in characterizing ice surface roughness by combining images from the 60° forward viewing and 60° aft viewing cameras from NASA's Multi-angle Imaging SpectroRadiometer (MISR). This angular reflectance composite formed the 'normalized difference angular index' (NDAI), which compares the relative amount of backward to forward scattering of sunlight. It has been shown that the NDAI is a qualitative proxy for surface roughness and can be used to discriminate between rough bare ice and snowcover, especially when paired with observational data acquired from aircraft over sites such as widely studied Jakobshavn Isbræ on the western margin of the Greenland ice sheet. In that work and here as well, roughness is defined as the root mean square (RMS) of deviations of measured surface elevations from a fitted plane of a specified extent, in this case an 80 m "platelet" from the Airborne Topographic Mapper (ATM) lidar instrument, as depicted in Figure 1.

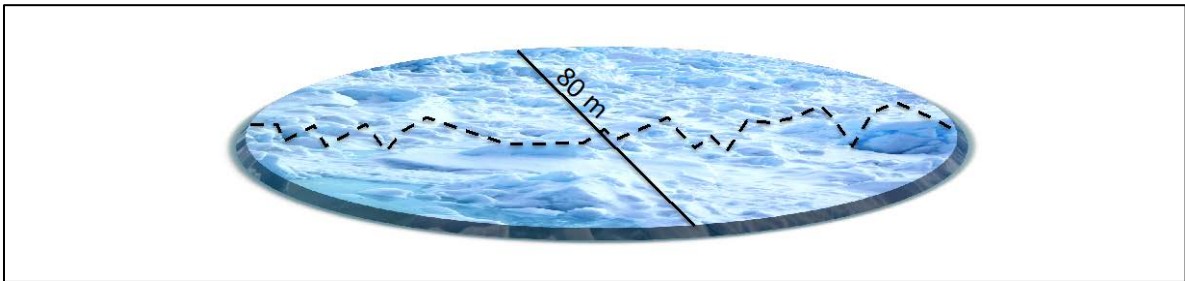

**Figure 1.** Depiction of sea ice surface roughness in an Airborne Topographic Mapper (ATM) 80 m platelet.

Although a positive correlation between NDAI and roughness clearly exists, this initial approach was fundamentally limited by its qualitative nature. While airborne lidar provides excellent spatial resolution and the ability to map centimeter to meter-scale changes in roughness, airborne data lack spatial coverage and regular repeat frequency. The advantage of employing a satellite-based retrieval of roughness is that it provides more complete coverage over remote sea ice regions and has the potential to map seasonal changes in sea ice roughness. Therefore, the overarching goal of this project was to develop a satellite-based technique for quantitative mapping of sea ice roughness with adequate quantitative precision to assess surface roughness characteristics across the physically relevant scales, from centimeters to meters, and with the potential for implementing the technique over large areas.

Previous satellite efforts to map sea ice surface roughness have employed passive microwave, radar altimetry, scatterometry, and laser altimetry. Passive microwave has a footprint of 6–25 km, so coverage is excellent, but the coarse spatial resolution doesn't provide process scale information. RADARSAT-2 ScanSAR Wide (SCWA) data have been compared with manual sea ice observations to

create maps of sea ice ridging classes at 500 m spatial resolution, but these are qualitative and currently limited to the Baltic Sea [3]. Cryosat-2 radar altimetry [21] offers higher spatial resolution (1650 m × 380 m), but surface roughness is determined by least-squares fitting of the radar return waveform [22] and can be confounded by backscatter angle. Scatterometry is sensitive to sea ice roughness though roughness is only one of several factors that simultaneously affect backscatter [23,24]. ICESat laser altimetry measures sea ice elevations with ellipse-shaped footprint of 57.3 m and 170 m spacing [25] so spatial coverage is not continuous; elevation retrievals are affected by roughness within each footprint so it remains challenging to provide process-scale information on sea ice roughness [26].

In the following sections, we outline a new method for quantitatively determining sea ice roughness using multi-angular reflectance data acquired by the Multi-angle Imaging SpectroRadiometer (MISR) and calibrated to quantitative values of roughness using airborne lidar data.

## 2. Methodology

### 2.1. Description of the Multi-angle Imaging SpectroRadiometer (MISR)

Launched on NASA's Terra satellite in 1999, MISR views the amount of sunlight scattered in 9 along-track directions and at 4 wavelengths (446 nm, 558 nm, 672 nm, and 867 nm) [27]. The near-polar, sun-synchronous orbit of MISR has an orbital repeat period of 16 days, while it takes 9 days to map Earth's surface up to a latitude of 83°. Each of MISR's 233 orbital paths contains a swath of data divided into 180 blocks. Each MISR image block is 563 km wide with the nadir camera filling 376 km and the off-nadir cameras filling 413 km; the remainder of the block contains fill values.

The 9 MISR cameras are labeled Df, Cf, Bf, Af, An, Aa, Ba, Ca, and Da. The initial letter (A, B, C, D) of each camera denotes the focal length and viewing angle of the camera. The viewing angles for the off-nadir A, B, C, and D cameras are 26.1°, 45.6°, 60.0°, and 70.5°, respectively, relative to the horizontal plane on the Earth's surface. The lowercase letters (f, n, a) denote whether the camera is looking forward, nadir, or aftward, respectively.

### 2.2. Calibrating and Mosaicking Multiangular Reflectance Data to Create Roughness Maps

The overall approach for creating quantitative maps of surface roughness entailed using multiangular reflectance and calibrating these data with measurements of ice sheet surface roughness from airborne LiDAR data. Calibration was achieved by developing an empirical relationship between three angular reflectance quantities (reflectance values from MISR Ca, Cf, and An cameras) and the ATM roughness data. These data were the basis for a four-dimensional (4-D) model that assigns roughness values to MISR data based on the empirical relationship between angular reflectance and measured ice sheet surface roughness. There are four main steps in developing maps of ice sheet surface roughness: (1) Preprocessing the data; (2) building the 4-D model; (3) applying the model to new MISR data; and (4) spatially aggregating and mosaicking the data to create roughness maps. We also assessed the model skill using a variety of metrics.

### 2.3. Data and Data Preprocessing

We began with MISR top-of-atmosphere (TOA) radiance data (ML1B2E), which has the nine MISR cameras geometrically rectified to the Earth ellipsoid. Based on previous work by Nolin et al. [20] and Nolin and Payne [28], we used the red band from the Ca, Cf, and An cameras to optically characterize the forward and backward scattering characteristics of the sea ice surface. The relative amount of backward scattering to forward scattering is an indication of the relative surface roughness.

To assign a quantitative roughness value to the MISR angular reflectance data, we required measurements of sea ice surface roughness over a wide range of roughness values and at a spatial scale that is comparable to the MISR 275-m pixel size. For this purpose, we used roughness data from the Airborne Topographic Mapper (ATM) IceBridge ATM L2 Icessn Elevation, Slope, and Roughness, Version 2 [29]. The ATM is a conical scanning airborne laser altimeter that measures the distance from

the aircraft to the topographic surface below using a 532 nm pulsed laser [30]. For this study, we used ATM data acquired with the instrument mounted on either the NASA P3 or DC8 aircraft. The instrument flies at a nominal altitude of 500–750 m, using a scanner angle of 22° with a laser footprint of ~1 m, a footprint spacing of ~3 m, horizontal accuracy of 0.74 m, and vertical precision of 3 cm [30–32]. Kurtz et al. [25] used ATM data to assess surface elevation estimates from ICESat. Here, we used the ATM icessn-processed data, which were resampled from the original high-volume elevation data set by fitting ~80-m diameter overlapping "platelets" along the flight line using along-track and cross-track slopes and then computing the average and root mean square (RMS) deviation of all elevation points within each platelet to yield values for average elevation and roughness, respectively. We used only the roughness values from the 0 degree (nadir) scan angle of the ATM.

Using the geolocation data and time tags from the ATM data, we identified the corresponding MISR image closest in time (±1 day) of the ATM overflight. The interval of ±1 day from the ATM flight path was used because possible shifting sea ice conditions could affect agreement between ATM and MISR sea ice roughness values. Since the ATM data acquisitions are flown under clear sky conditions, the corresponding MISR reflectance data must also be cloud-free. Extending the data acquisition date by ±1 day helps identify cloud-free MISR data that are near-coincident with the ATM data. To identify and remove MISR reflectance data that are contaminated by clouds, we used the MISR Angular Signature Cloud Mask (ASCM, MIL2TCCL) product [33]. Specifically, we used the "high confidence clear" data from this cloud mask product. We note that the cloud masks over sea ice tended to exclude smooth snow-covered ice because of the similarity to clouds. Thus, for the most accurate calibration we relied on a combination of the cloud mask and careful visual checks. If both the visual check and the ASCM indicated the presence of a cloud, we assumed the area was cloudy. If the ASCM showed cloudy conditions but the visual check showed it was clear, then we assumed it was clear. If cloudy conditions were determined to exist, then we excluded both the MISR reflectance and the ATM surface roughness data for those locations. There were no cases where the ASCM showed clear conditions and the visual check determined cloudy conditions.

Because the MISR footprint is larger than that of the ATM icessn-processed roughness data, there were multiple ATM roughness values within each MISR pixel. The latitude and longitude of each ATM roughness value was converted to the center latitude/longitude value of the closest MISR pixel. The ATM roughness values within each corresponding MISR pixel were then averaged, and we recorded the average roughness, standard deviation, and number of ATM values. The average and number of ATM values were later used in a weighting function described in the following section.

### 2.4. Building the 4-D Surface Roughness Model

In the roughness model, each set of three MISR reflectance values (from the Ca, Cf, and An cameras) has a corresponding ATM-derived roughness value. We refer to this as a four-dimensional (4-D) model. In constructing the empirical relationship between the angular reflectance values and lidar-derived surface roughness values, we began by compositing the ATM icessn-processed roughness values located within a MISR pixel and assigning the average to the MISR pixel. The average, standard deviation, and number of ATM roughness values used in the calculation were retained for subsequent use. This average roughness was then the modeled roughness for the MISR pixel.

We used a nearest-neighbor approach to assign a surface roughness value for each triad of Ca, Cf, and An reflectance values. Thus, we needed to determine the prediction radius within which one or more ATM roughness values were found. Using all the 2013 and 2016 ATM roughness data and corresponding MISR data (17,250 pixels and 19,425 pixels, respectively), we computed an optimal prediction radius, $r_{pred}$. We varied $r_{pred}$ to determine whether an optimal $r_{pred}$ value that gives the highest correlation with the "true" roughness (as measured by the ATM) exists. Note that not all MISR-ATM model data points will have surrounding data points within the prediction radius. As $r_{pred}$ is increased, there is a greater chance that there will be an ATM roughness within the search radius, but there is also a greater chance that the predicted roughness will have a lower correlation with the

actual roughness of the point to be modeled. So, another important parameter is the coverage, C, of the ATM model as a function of the prediction radius. Coverage is defined as the number of MISR pixels within the prediction radius that can be assigned a roughness value ($n_{pred}$) relative to the total number of valid MISR pixels in the image ($N_{MISR}$):

$$ C = \frac{n_{pred}\left(r_{pred}\right)}{N_{MISR}} \tag{1} $$

It is desirable to maximize C, but as the prediction radius is increased, there is a greater chance that the predicted roughness will have a lower correlation with the actual roughness of the point to be modeled. This is important because for nearest neighbor classification, a small radius could result in no corresponding ATM roughness value (no roughness value assigned to a MISR pixel) while a large prediction radius could contain widely varying ATM roughness values (inaccurate roughness value assigned to a MISR pixel).

We computed the average of the ATM roughness values within each progressively larger concentric sphere and then computed the correlation between the "true" ATM-measured roughness value and the estimated roughness value for each value of $r_{pred}$. This was performed for each of the >1,000,000 pixels used to generate the 4-D model. The optimal radius was determined to be 0.025. This is a dimensionless value since it is derived from surface reflectance.

Using the 0.025 value for $r_{pred}$, we then assign a roughness value to each new MISR pixel. The nearest-neighbor method was used to find those roughness values in the model data cloud closest to that of the new MISR pixel. The location of each point within the local prediction radius is determined by the following constraint:

$$ r_{pred}\sqrt{\left(Ca_m - Ca_p\right)^2 + \left(Cf_m - Cf_p\right)^2 + \left(An_m - An_p\right)^2} \tag{2} $$

where, $Ca_m$, $Cf_m$, and $An_m$ are the x, y, and z coordinates of the model data point, and $Ca_p$, $Cf_p$, and $An_p$ are the x, y, and z coordinates of the prediction data point. Because in creating the model we averaged the multiple ATM roughness values within a MISR pixel, not all model points were created with the same number of ATM roughness values. To account for this, a weighting scheme was used that is based on the number of ATM roughness values that comprised each point in the data model. The weighted average roughness value ($\mu_w$) of the prediction data point is as follows:

$$ \mu_w = \frac{\sum_{i=1}^{m} n_i x_i}{\sum_{i=1}^{m} n_i} \tag{3} $$

where, $\mu_w$. is the weighted average roughness predicted for the new MISR pixel, $m$ is the total number of model data points within $r_{pred}$, $n_i$ is the number of ATM roughness values used to compute the roughness for that model data point, and $x_i$ is the model roughness values within $r_{pred}$.

### 2.5. Applying the Model to New MISR Data

Next, the 4-D model was used to assign an ATM roughness value to every pixel in a new MISR sea ice image. Preprocessing of the MISR data is the same as with the MISR data used to construct the 4-D model: Converting TOA radiance (from the Ca, Cf, and An cameras) to reflectance. The reflectance values were placed into the 4-D model space and, using $r_{pred}$ = 0.025, each MISR roughness estimate was formulated as the weighted average of points within $r_{pred}$.

## 3. Results

### *3.1. Roughness Maps from MISR*

The 4-D model described above was used to convert MISR reflectance data for all cloud-free MISR orbits during April–July 2013 and 2016 in the Arctic region extending from the Beaufort Sea to Ellesmere Island in the Canadian Archipelago. This region was selected because (a) prior research showed this to be an important region of convergence and divergence [34] and (b) ATM data were available for the region so we could calibrate and validate the MISR roughness mapping model there. Figure 2 shows the results of the MISR-estimated surface roughness algorithm for a spring image (26 April 2016) and a summer image (15 July 2016). The spring image is almost entirely cloud-free. It shows an extensive area of relatively smooth sea ice with roughness values of mostly 12–20 cm. There is one very rough area of ice in the Greely Fiord of Ellesmere Island where there are ice roughness values exceeding 80 cm. In this region, the ice pack represents dynamic glacier ice that has flowed into the fiord and frozen in place. The summer image contains clouds in the center portion of the image. Focusing on the cloud-free southern part of the image the roughness image shows ice floes, ridges, cracks and other convergence-divergence features. As in the spring image, the largest values for ice roughness were found in Greely Fiord.

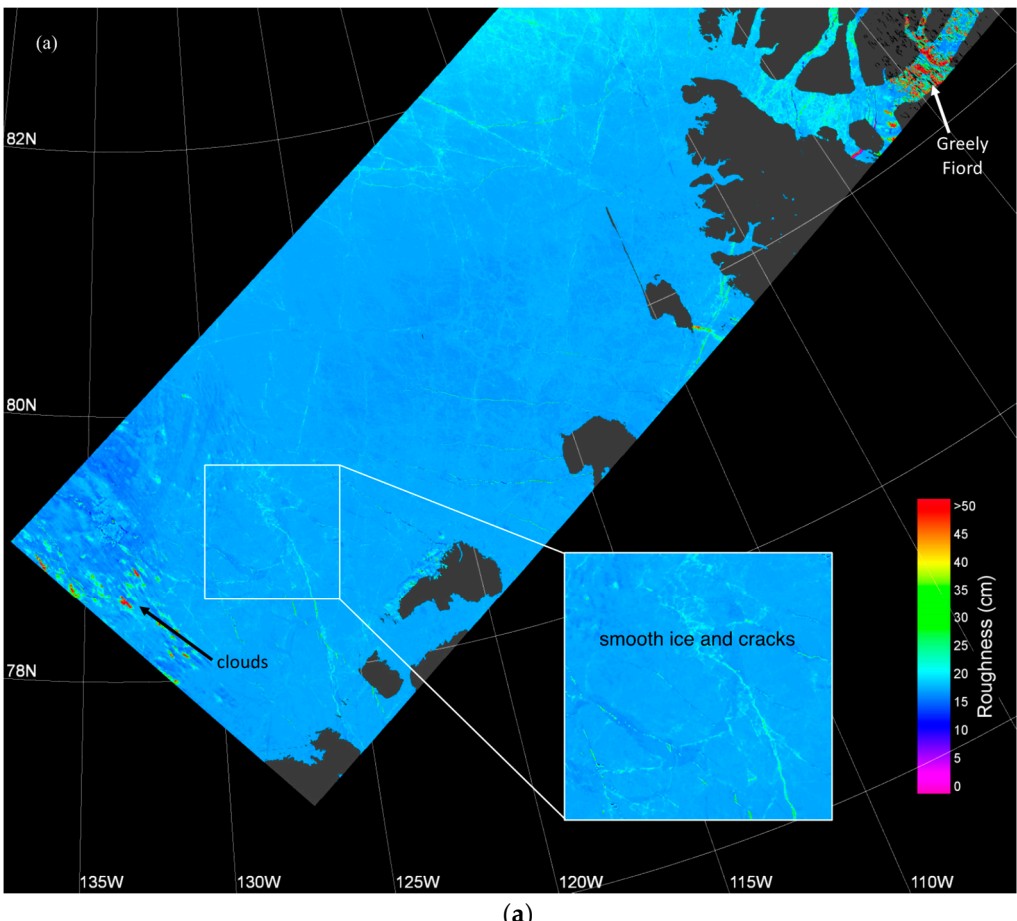

(**a**)

**Figure 2.** *Cont*.

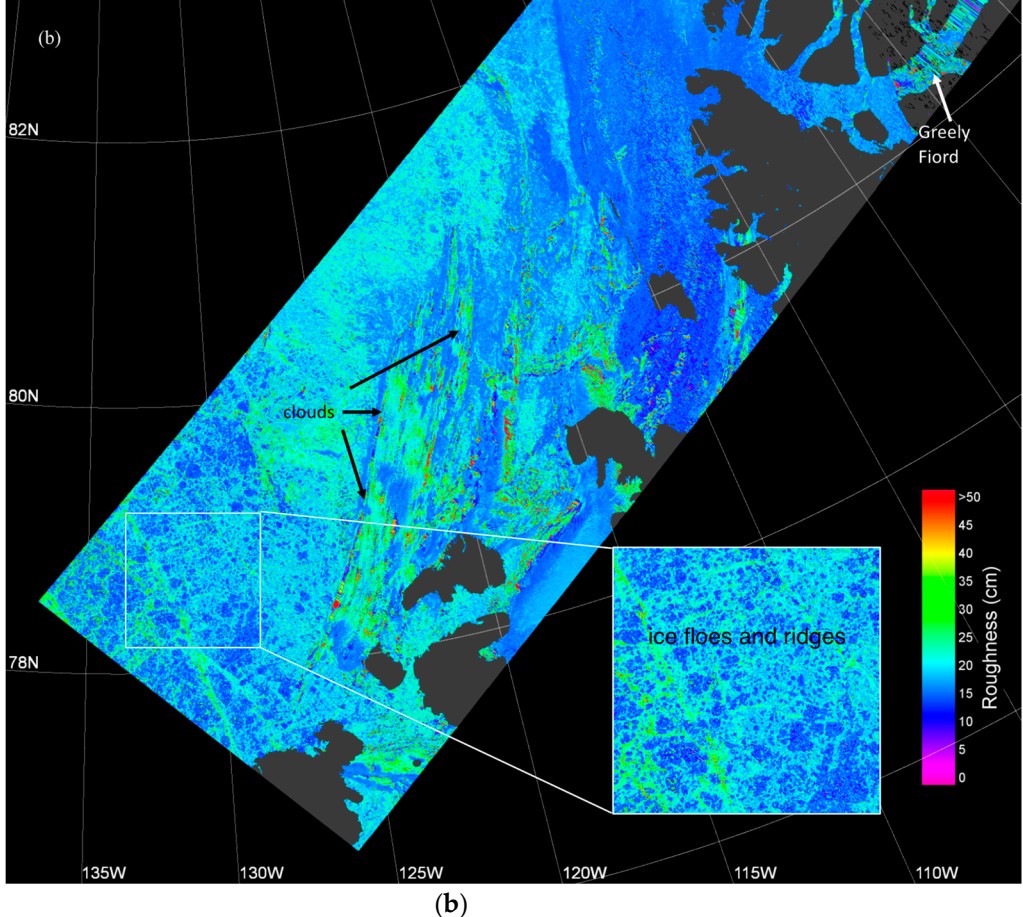

**(b)**

**Figure 2.** Multi-angle Imaging SpectroRadiometer (MISR)-estimated sea ice surface roughness for (**a**) 26 April 2016 and (**b**) 15 July 2016. The Beaufort Sea region has been highlighted to show the seasonal changes in roughness. Clouds have not been masked and appear as rough streaks. The roughest ice in both images is in the Greely Fiord of Ellesmere Island (upper right of both images), where ice convergence creates large ridges.

*3.2. Assessment of Results*

To assess the accuracy of the method, MISR-estimated sea ice roughness values were compared with corresponding independent ATM roughness data, which had not been used to construct the 4-D model. Results were assessed for three categories: smooth (roughness < 20cm), rough (roughness < 100 cm), and smooth-rough transitions. Figure 3a–c shows MISR-estimated and ATM roughness values from a validation data set that used co-located MISR and ATM roughness data. As before, the MISR data acquisition was within ±1 day of the ATM overflight. The 80 m ATM roughness values within each MISR pixel were averaged so that an individual ATM and MISR value could be compared. The validation data were from ATM overflight data during April–May 2016 and were selected to encompass smooth, rough, and smooth-rough transitions. Figure 3 and Table 1 show that, for smooth ice, the mean values were virtually identical, and the range and variance were also quite similar between MISR-estimated and ATM roughness. This was not the case for rough ice where the mean values were close, but the ATM roughness values had a much greater variance than the MISR-estimated roughness values. The coefficient of variation ($R^2$) for MISR-estimated vs. ATM roughness values was 0.52 for smooth sea ice areas and 0.39 for the rough sea ice areas. Results for smooth-rough transitions show that when there are abrupt spatial changes in roughness, the MISR-estimated roughness values closely match the spatial changes of ATM roughness. $R^2$ values for spatial transitions are not relevant and are not reported.

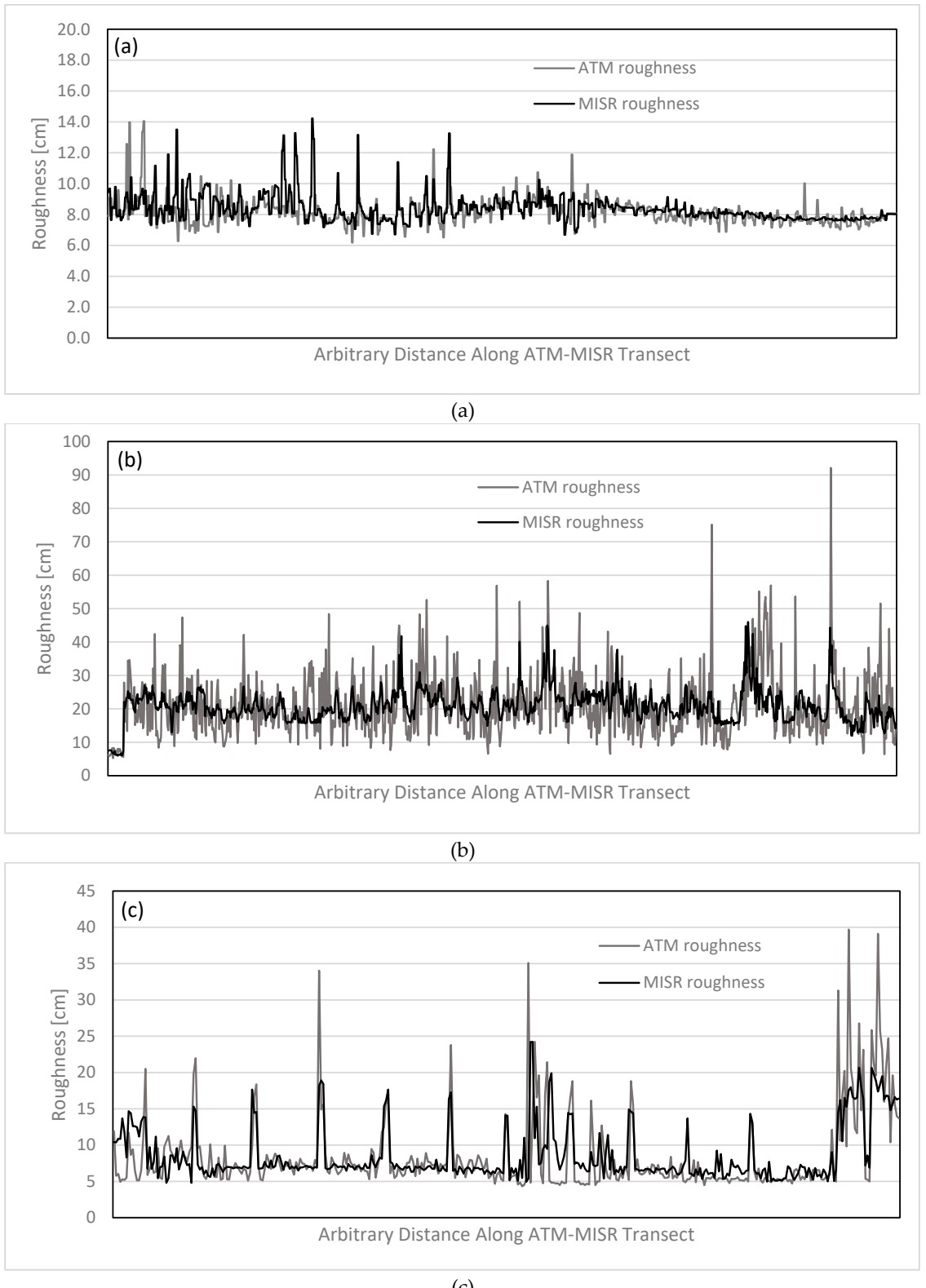

(a)

(b)

(c)

**Figure 3.** Comparison of ATM and MISR-estimated sea ice surface roughness from April and May 2016. (**a**) Smooth (<20 cm roughness), (**b**) rough (<100 cm roughness), and (**c**) smooth-rough transitions.

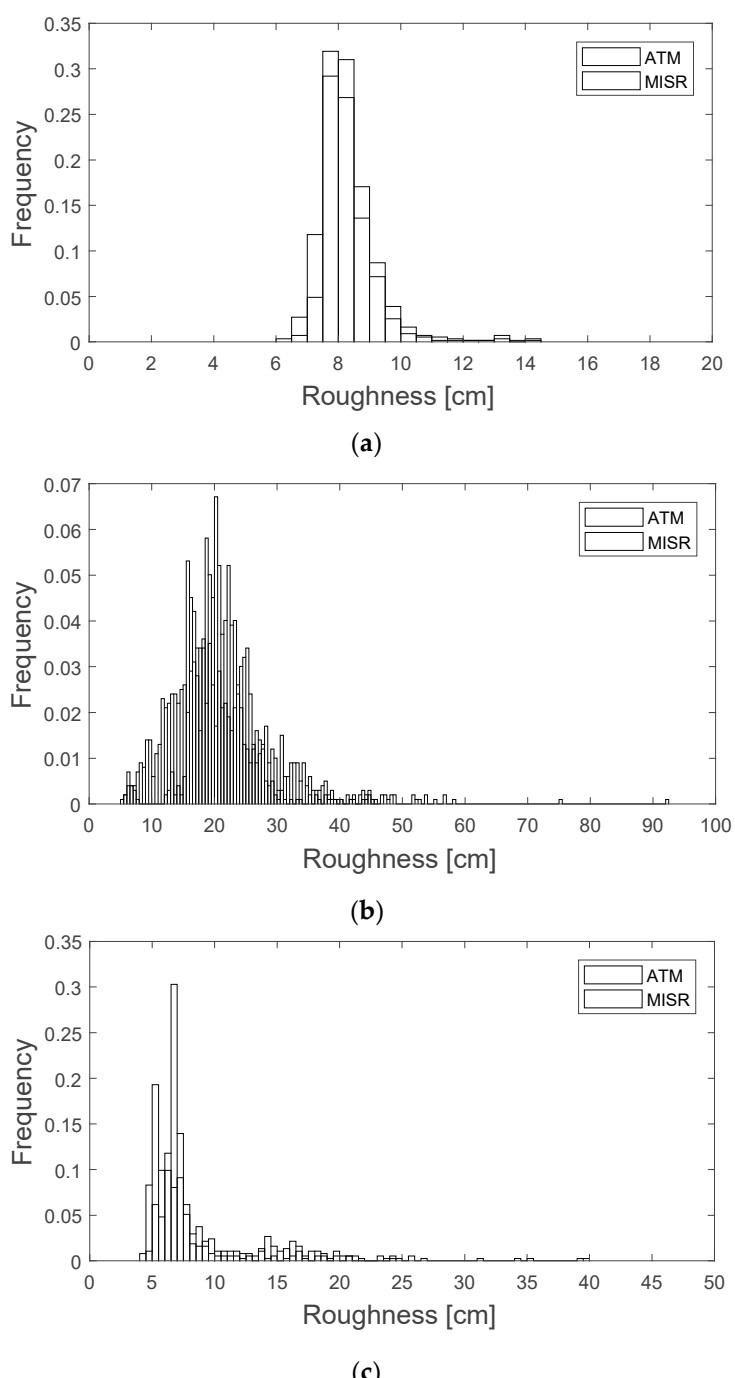

**Figure 4.** Histograms of ATM and MISR roughness frequency distributions for sea ice areas that are (**a**) smooth (<20 cm roughness), (**b**) rough (<100 cm roughness), and (**c**) smooth-rough transitions.

**Table 1.** ATM and MISR-estimated roughness statistics for the histogram data in Figure 4a,b.

| Roughness Type | Instrument | Mean Roughness [cm] | Median Roughness [cm] | Variance [cm²] | Maximum Roughness [cm] | Minimum Roughness [cm] |
|---|---|---|---|---|---|---|
| Smooth | ATM | 8.2 | 8.0 | 0.9 | 14.1 | 6.2 |
| | MISR | 8.4 | 8.1 | 0.9 | 14.2 | 6.7 |
| Rough | ATM | 21.2 | 19.4 | 86.0 | 92.1 | 5.3 |
| | MISR | 20.8 | 20.4 | 23.1 | 46.0 | 6.0 |

*3.3. Frequency Distributions*

A comparison of the frequency distributions of ATM and MISR-estimated roughness values indicated that the ability of this method to accurately represent sea ice roughness depends on the magnitude of roughness. We note that the values shown here are separate from the ATM and MISR data that were used to construct the 4-D model used to estimated MISR roughness. The frequency distributions shown in Figure 4 demonstrate these differences for smooth, rough, and smooth-rough transitions. The mean and median of ATM and MISR roughness values were nearly identical for smooth sea ice. For rough ice (Figure 4b), the mean and median of ATM and MISR roughness values were also quite similar, but the range and variances were very different. The histogram in Figure 4c shows the characteristic bimodal distribution for the smooth and rough sea ice areas. In the smooth-rough transition data, most of the areas are smooth (roughness of about 7 cm) with short transitions to rougher ice (roughness of about 17–35 cm). The much smaller secondary mode Figure 4c shows this for both ATM and MISR-estimated roughness values.

## 4. Discussion

In the context of other methods to estimate roughness, this work stands out because it provides spatially extensive roughness information at a horizontal scale of 275 m and with vertical precision of centimeters to decimeters. This method is effective for quantifying and mapping the surface roughness of smooth ice (corresponding to thinner first year sea ice), rough ice (corresponding to thicker first year, multi-year sea ice, and heavily deformed ice within fjords), and ice with spatial roughness transitions between smooth and rough (corresponding to ice floes, surrounded by ridges and leads).

The MISR lower range of roughness values and smaller variance compared with ATM data is likely due to the differences in spatial resolution between MISR and ATM. Finer scale roughness features that appear in the ATM data are not resolved in a MISR pixel.

There are several sources of error to consider. First, the potential temporal offset of $\pm 1$ day between ATM data acquisition and a MISR overpass may result in a co-location error: A MISR pixel may not represent the same ice conditions as the ATM footprint. This type of error is relatively small in spring within the Canadian Arctic because cold temperatures tend to reduce ice motion at that time of year. Regions and times of year where sea ice velocity exceeds MISR pixel size may require daily concurrence between MISR and ATM overflights. A second source of error is the time period used in the calibration. If the 4-D model is constructed using data from April–May, it may omit ice characteristics that are present later in the summer. Most ATM overflights occur in spring, and it would be helpful to have a larger number of flights in summer and fall. Additional errors may enter from the 4-D modeling process. Calculation of the prediction radius is optimized. but tests show that even when the prediction radius is varied between 0.020 and 0.030, the optimization results do not change significantly; thus, small changes in $r_{pred}$ will not contribute much to the full error budget. Moreover, the relatively low sensitivity of the results to small changes in $r_{pred}$ suggest that this parameter is transferable to other sea ice regions beyond what was tested in this study. The largest challenge in the modeling process is identifying cloud-free MISR imagery. Currently, there is no cloud mask that effectively identifies clouds over sea ice that can be used in an automated manner. Visual identification of clouds remains the best approach but is overly time consuming.

## 5. Conclusions

This multi-angular remote sensing approach to mapping sea ice surface roughness appears to show promise. These results demonstrate the ability to map changes in surface roughness at centimeter to decimeter scales in a 275 m MISR pixel. MISR estimates of surface roughness suggest the ability to map features such as cracks, ridges, smooth snow, frozen leads, and meltponds. While this method is limited to cloud-free images during the sunlit season, it has the potential to map sea ice roughness

during the critical spring season prior to meltpond formation. Calibration using the ATM lidar provides the key data to convert multi-angular reflectance to quantifiable roughness estimates.

With further analysis, this method could be used to map ice types based on roughness characteristics, to examine changes in the extent of first year and multiyear ice, and to map areas where leads are present. One could also analyze spatial patterns of sea ice roughness using frequency analysis to better characterize convergence and divergence features. It is hoped that this approach can also be combined with new data to better train and constrain MISR-estimated sea ice roughness. Future work should apply this method to a much larger region of the Arctic and over the full MISR record from February 2000 to the present. These sea ice roughness data can improve interpretation of sea ice thickness from spaceborne radar and laser altimeter instruments. Applying this new roughness method to the entire Arctic will benefit the Arctic sea ice science community and has potential for operational use in Arctic navigation.

**Author Contributions:** Conceptualization, A.N.; methodology, A.N. and E.M.; programming, E.M.; validation, E.M. and A.N; formal analysis, A.N. and E.M.; investigation, A.N.; resources, A.N.; data curation, A.N.; writing—original draft preparation, A.N.; writing—review and editing, A.N. and E.M.; visualization, A.N. and E.M.; supervision, A.N.; project administration, A.N.; funding acquisition, A.N.

**Funding:** This research was funded by the NASA/Jet Propulsion Laboratory Contract #1540933.

**Acknowledgments:** The MISR team and Langley Atmospheric Science Data Center are acknowledged for their assistance in procuring data.

**Conflicts of Interest:** The authors declare no conflict of interest. Any brand name or companies in the paper are not endorsements. The funding sponsors had no role in the design of the study; in the collection, analyses, or interpretation of data; in the writing of the manuscript; and in the decision to publish the results.

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
