# Peer review of "Arctic Sea Ice Surface Roughness Estimated from Multi-Angular Reflectance Satellite Imagery"

_remotesensing, doi:10.3390/rs11010050_

Round 1

Reviewer 1 Report

The work presented here is well written and clear. The potential of using pan-Arctic available data is very promising. 

Major comments

At the moment the connection between different ice types such as; first year ice and multi-year ice to the roughness information extracted within this study are introduced first in the discussion. Moving this connection to earlier in the manuscript may make it easier to justify the importance of the work, e.g. multi-year ice and deformed sea ice is harder to break for ice breakers than first year ice. There is also substantial work done connecting sea ice type and roughness e.g. lately work by Gegiuc et al. (2018). 

Figure 4. There seems to be some overlap between the different sea ice roughness, where e.g. the smooth-rough ice mostly overlaps the smooth ice. Further on line 207 smooth ice is defined as <20 cm, from figure 4b this includes nearly half of the rough sea ice. What is the reason for this? A discussion around this is missing from the manuscript. It would also be good if Figure 4b says that the rough sea ice is sea ice with roughness’s from 20-2500px. 

The Canadian Arctic has less sea ice drift than for example the European and Russian Arctic sea ice. The ±1 day in between the ATM data and the MISR used in the study here would need to be adjusted to work in those regions. It would be nice to see this specified within the manuscript. 

Minor comments

Line 10: The reference to «we» is a bit strange considering that this is a one-author manuscript. 

Line 30: What is the meaning of skilful here? 

Line 38. Please replace [ref] with the correct reference. 

Figure 1. Please replace the “?” 

Line 147. What is the unit for the optimal prediction radius? 

Line 155 and 157. Please be consistent with npred or npre.

Line 302. Please quantify “a little”. 

References

Gegiuc, A., Similä, M., Karvonen, J., Lensu, M., Mäkynen, M., and Vainio, J.: Estimation of degree of sea ice ridging based on dual-polarized C-band SAR data, The Cryosphere, 12, 343-364, https://doi.org/10.5194/tc-12-343-2018, 2018.

Author Response

Please see the attached pdf file. Comments from Reviewer 1 are in black. My responses are in red.

Reviewer 2 Report

The manuscript seeks to establish a four-dimensional model to retrieve ice surface roughness by using data from multi-angle imaging spectroradiometer. Although I appreciate the importance of the problem, this manuscript has major issues that lead me not to recommend it for publication in its current form.

General comments:

G1: First, and most importantly, the lidar-derived roughness measurements from the ATM were taken as the "true" values and reference for the empirical model developed. However, the reliability of the lidar-derived roughness itself is doubtful. Thus, the performance of the proposed method is dubious.

G2: The application of the model was conducted on only two MISR images. If the writers wish to demonstrate its ability compared with ATM, then more experiments should be done.

G3: Concerning the setting of parameter rpred: is the optimal rpred feasible in other studies? What factors affect this parameter? Is it reasonable to use the same rpred for images with different cloud conditions?

Line-by-line comments:

L1: Line 23: I would suggest some references to be provided for the first sentence.

L2: Line 38: [ref] please check.

L3: Figure 1: there are two interrogation marks.

L4: Line 79: the specific wavelengths should be provided.

Author Response

Please see the attached file. Comments from Reviewer 2 are in black and my responses are in red.

Round 2

Reviewer 1 Report

The authors have satisfactory responded to all my comments. I therefore recommend the manuscript for publication. 

Author Response

Please see the attached file for my responses to the guest editors comments.
